# Walkability under Climate Pressure: Application to Three UNESCO World Heritage Cities in Central Spain

**Javier Velázquez** [1,*], **Javier Infante** [1], **Inmaculada Gómez** [1], **Ana Hernando** [2], **Derya Gülçin** [3], **Fernando Herráez** [1], **Víctor Rincón** [4] and **Rui Alexandre Castanho** [5,6,7]

1   Faculty of Sciences and Arts, Department of Environment and Agroforestry, Catholic University of Ávila, 05005 Ávila, Spain; javier.infante@ucavila.es (J.I.); inmaculada.gomez@ucavila.es (I.G.); fernando.herraez@ucavila.es (F.H.)
2   Silvanet Research Group, Universidad Politécnica de Madrid, Ciudad Universitaria s/n, 28040 Madrid, Spain; ana.hernando@upm.es
3   Faculty of Agriculture, Department of Landscape Architecture, Aydın Adnan Menderes University, Aydın 09100, Turkey; derya.yazgi@adu.edu.tr
4   Faculty of Pharmacy, Department of Pharmacology, Complutense University of Madrid, Plaza de Ramón y Cajal, s/n, 28040 Madrid, Spain; virincon@ucm.es
5   Faculty of Applied Sciences, WSB University, 41-300 Dąbrowa Górnicza, Poland; acastanho@wsb.edu.pl
6   CITUR–Madeira–Centre for Tourism Research, Development and Innovation, 9000-082 Funchal-Madeira, Portugal
7   College of Business and Economics, University of Johannesburg, Auckland Park P.O. Box 524, South Africa
*   Correspondence: javier.velazquez@ucavila.es

**Abstract:** Walkability is a modern concept that has become important in recent years due to the doubtless effects it has on aspects such as health and wellbeing, sustainable development, climate change, and tourism. It is necessary, therefore, that urban development strategies aim to achieve walkable cities. The main objective of this study is to define a methodology to calculate the walkability index in tourist cities and to predict the effects of climate change on this index, which is applied to three World Heritage cities in central Spain: Salamanca, Ávila, and Segovia. The methodology is developed in three phases. Phase I focus on the calculation of walkability, considering the following factors: facilities and services, accessibility, sidewalk width, population density, green areas, and urban trees. In Phase II, walkability in 2020, climate-related variables were added to the previous result: temperatures, solar radiation, and shadows. Finally, the third phase, walkability under climate change pressure in 2030, 2050, and 2100, establish predictions for different climate scenarios. The results show excellent walkability indices (higher) in city centers and newly built neighborhoods and low values in the rest of the peripheral areas, industrial estates, and neighborhoods. Climate predictions showed a generalized decrease in walkability over time, even higher in the scenario with high greenhouse gas emissions. Likewise, the models can be an excellent tool for the tourist management of cities since they show the most walkable areas and, therefore, the most suitable for tourist routes.

**Keywords:** walkability; climate change; tourism; geographic information system (GIS); remote sensing

## 1. Introduction

Walkability can be defined as the aptitude of the built environment to support and encourage walking by providing comfort and safety for pedestrians, connecting people to various destinations in reasonable time and effort, and providing visual interest in journeys along with the network [1]. In recent years, the concept of walkability has become relevant, both in popular and academic fields, as well as an aspect to consider in urban development [2,3]. There is a lack of consensus when characterizing and evaluating this concept since the variables that influence it are multiple and diverse [4,5]. However, there are numerous proposals. Ewing and Handy [6] consider that measuring walkability is

possible by using ratings from an expert panel. Bhattacharyya and Mitra [7] highlight a walking audit tool called PERS (Pedestrian Environment Review System), used mainly in the UK, and a walkability index known as Walk Score, which consists of an algorithm based on the distance between different services of various categories, but it has the drawback that it does not consider other factors such as the availability and width of sidewalks, the safety of the area or the topography. Lee and Talen [8] propose a hybrid methodology that combines modeling by using geographic information systems with the walker's perspective on Google Street View.

However, despite the complexity of defining and measuring the concept, its advantages are evident. Therefore, the future of the structure of the cities must be focused on creating walkable cities [9,10]. Speck [11] classifies the benefits of walkability into three groups: health, sustainability, and economy. The relationship between health and walkability is obvious since walking implies physical activity, which has important benefits for people [12,13]. Among others, it is possible to highlight the contribution to greater social interaction and an improvement of mental health [14,15]. In addition, there is much research literature that refers to the benefits of walkability in the treatment or prevention of specific diseases. Auchincloss et al. [16] conclude a factual reduction in both the incidence and severity of type 2 diabetes cases in walkable cities. Moreover, Steell et al. [17] link walkability with a decrease in obesity cases, and Celis-Morales et al. [18] argue that walking leads to a reduction in cardiovascular diseases and a lower incidence of different types of cancer.

Achieving more walkable cities is also an important measure to steer the urban structure in the direction of sustainable development [11,19]. In this sense, reducing dependence on the use of vehicles can lead to a decrease in air pollution and noise, a reduction in greenhouse gas emissions, lower traffic density, and more efficient use of urban land due to the reduction of parking spaces [20]. A city with wide walkable areas encourages greater equity among different social classes, ethnicities, sexes, and age groups [21,22]. Regarding the economic factor, walkability has a significant influence on tourist inflow because the best way to enjoy and get to know a city is by walking [23]. Moreover, if tourists can tour the city on foot, the subjective perception they will have after the visit will be very favorable [24]. Therefore, this fact is of utmost importance in localities whose economic engine is tourism, such as World Heritage cities, as well as in other historic cities with similar characteristics [25]. These cities included in the United Nations Educational, Scientific, and Cultural Organization (UNESCO) World Heritage List are a reference point for urban tourism, as they are towns that stand out for the exceptional cultural and historical values of their urban areas [26].

Hernández et al. [26] state that, due to the socioeconomic importance of tourism, it is necessary to plan and manage to ensure the sustainability and competitiveness of this activity. Thus, it is necessary to know as much as possible about the factors that influence the tourism sector of a city, and one of them is undoubtedly walkability, which depends on other aspects such as topography, climate, safety, aesthetic values, street organization, accessibility of destinations, greenery, maintenance, and public amenities [27–30].

Climate, among the above-mentioned factors influencing walkability, is changing globally, and urban planning will have to consider the effects of this climatic alteration [31–33]. These consequences are difficult to predict, but the current scenario and research show some of them. Precipitation increases its intensity while reducing its frequency (number of rainy days), leading to a significant increase in water erosion [34]. Sea levels are rising, which causes damage to coastal infrastructures and huge population displacements [35]. Droughts will be more frequent and severe [36]. Soil salinization due to higher evapotranspiration rates will become widespread with consequent impacts on primary production [37]. Finally, although many more effects could be cited, heat and cold waves will become more frequent and intense [38]. Most of these consequences will have negative effects on walkability since the predisposition of dwellers to walk is considerably reduced by adverse weather, either due to heavy rainfall or broiling sun [7]. Therefore, it is necessary for cities to adapt to this

new climate scenario. As already mentioned, this adaptation is especially important in those whose economic engine is tourism, as is the case in cities included in the UNESCO World Heritage List, as well as in others with a large influx of tourists.

While previous works have focused on walkability from the perspective of citizens' wellbeing (e.g., [39]) or considering walkability and tourism (e.g., [40]), this study shows a potential combination of the walkability index within the tourism context of three World Heritage cities, linking not only walkability and tourism but also the potential affection due to climate change trends. Nevertheless, further studies are required to sustain this hypothesis with more reliability. The main objective of this work is to propose an innovative methodology for calculating a walkability index applicable to tourist cities in order to address the issues outlined above. It also presents the current climate scenario and predicts three future scenarios, 2030, 2050, and 2100, to provide a tool to facilitate tourism management in adapting to climate change.

## 2. Material and Methods

### 2.1. Study Area

The study was carried out in three Spanish cities of Castilla y León included in the World Heritage List of UNESCO: Ávila (1985), Segovia (1985), and Salamanca (1988). These cities are quite close to the geographical center of the Iberian Peninsula, in the South of the Northern subplateau (Figure 1), a Cenozoic basin drained by the Duero River and its tributaries. This depression is one of the highest intramountain basins in Europe, an aspect that influences its climatic characteristics [41]. The municipality of Ávila covers an area of 231.9 km$^2$ and has a population of 58,369 inhabitants [42], which means an average population density of 251.7 inhabitants/km$^2$. It is located 1132 m above sea level and is the highest capital of a province in Spain. The municipality of Salamanca occupies an area of 39.34 km$^2$ and houses a population of 144,825 inhabitants [42], which means an average population density of 3751.9 inhabitants/km$^2$. The city is located at an altitude of 802 m above sea level. The municipality of Segovia occupies an area of 163.59 km$^2$ and has a population of 52,057 inhabitants [42], which means an average population density of 318.2 inhabitants/km$^2$. This city is located at an altitude of 1005 m above sea level.

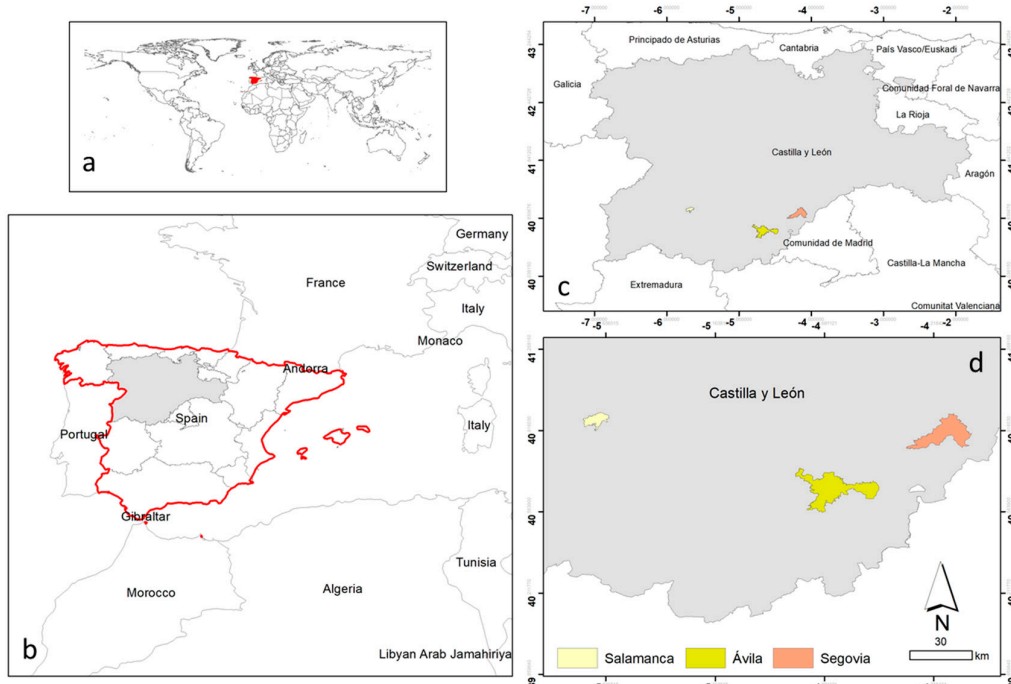

**Figure 1.** (**a**) Location of Spain, (**b**) Location of the Autonomous Communities in Spain, (**c**) Location of the Autonomous Community of Castilla y León, (**d**) Location of the study area in Castilla y León.

According to the updated Köppen-Geiger classification, Ávila, Salamanca, and Segovia are included in an area classified as a Csb climate (temperate climate with dry and mild summer) [43]. The average annual temperature of this climate is 11.5 °C, and the total annual precipitation is 424 mm (Climate-Data, n.d.). These three cities have been chosen because tourism is their economic engine [44], and therefore, it is essential to know which areas are the most walkable for the implementation of tourist routes and for the installation of commercial premises. In addition, their location in the Duero basin makes them particularly susceptible to climate change since significant changes are foreseeable in this basin, such as an increase in average temperature, a higher proportion of warm nights and days, and more widespread heat waves, as well as a trend towards a decrease in precipitation and in rainy days with an increase in the length of the dry period [45].

### 2.2. Methodology

This paper proposes a methodology applicable to tourist cities to calculate a walkability index for current and future scenarios. This methodology has three consecutive phases, and its implementation is executed with geographic databases and Geographic Information Systems software. The software used has been QGIS, with the QuickOSM add-on (QuickOSM, n.d.), using OpenStreetMap (OSM). The purpose of Phase I is to calculate a current walkability index for the cities of Ávila, Salamanca, and Segovia. In this phase, modeling has been performed with the following variables: facilities and services, access, sidewalk width, population density, green areas, and urban trees. Phase II consists of incorporating three climate-related variables (temperature, solar radiation, and shadow) into the walkability index calculated in the previous phase. Due to the climate in the region (continental dry) and the high tourist season during late spring and summer, those climate-related variables have been considered to be the ones more affecting of walking comfort [46]. The climate data for the three municipalities are updated up to the year 2020. In Phase III, a study of the walkability index under future climate pressure in 2030, 2050, and 2100 is performed. Finally, Phase II and Phase III are compared. Figure 2 shows an outline of the developed process.

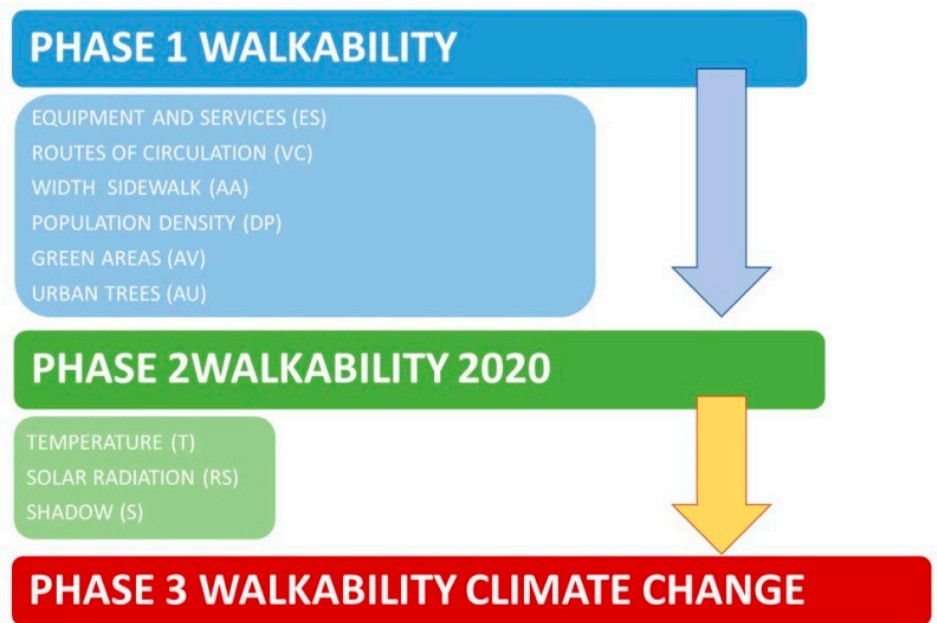

**Figure 2.** Used methodology—diagram.

### 2.2.1. Phase I. Walkability

The urban model should include green spaces, facilities, and services that should be accessible and comfortable, and the time spent walking between them should be as short as possible. Considering the ideal city described by Lopera [47], the following variables are proposed to calculate the current walkability: facilities and services, accessibility, sidewalk width, population density, green areas, and urban trees. To quantify these variables, this research uses the proposal of Rattan [48] and the equations proposed by Gutiérrez-López et al. [49]. The study is undertaken for the entire municipalities, but special attention is paid to the old downtown and the neighborhoods with the greatest influx of visitors.

Variable 1. Facilities and Services: they are a key element in urban planning since they are necessary for proper livability and the promotion of tourism [50]. The analysis of pedestrian intensity is based on the facilities and services that are located within the study area, represented as "FS". The location of facilities is obtained from the OMS 'amenity' file, while services are obtained from the 'leisure' file. It includes supermarkets, shops, public transport, bars and restaurants, primary schools, banks, pharmacies, and medical services. To calculate the percentage of proximity to these facilities and services (Equation (1)) is necessary to obtain the distance, represented as "A".

$$FS = \frac{\text{Surface area occupied by facilities and services in A } (m^2)}{\text{Surface area of A } (m^2)} \cdot 100 = \% \text{ facilities and services} \qquad (1)$$

Variable 2. Accessibility: based on infrastructure availability. In this sense, deployment and maintenance of streets and roads are essential since only in this way can a city guarantee the essential needs of the population [51,52]. The analysis of pedestrian intensity is based on the streets within the study area, which is represented as "AC". To calculate this AC, Equation (2) has been used, where the presence of roads, pedestrian paths, or other connection infrastructures (such as stairs) are considered.

$$AC = \frac{\text{Surface area occupied by access in A } (m^2)}{\text{Surface area of A } (m^2)} \cdot 100 = \% \text{ access} \qquad (2)$$

Variable 3. Sidewalk: it is a fundamental variable for urban development, social integration, and safety [42]. The usable width of a sidewalk is the space that can be effectively used by pedestrians in their movements. Analysis of pedestrian intensity is based on the sidewalk width, which is represented as "SD", which is calculated with Equation (3):

$$SD = \frac{\text{Surface area occupied by sidewalks in A } (m^2)}{\text{Surface area of A } (m^2)} \cdot 100 = \% \text{ sidewalk width} \qquad (3)$$

Variable 4. Population Density: it is one of the most influential factors in walkability [3,4]. The data for this analysis were obtained from the website of the Spanish National Statistics Institute in its Census Areas section (INE, 2020). To obtain the population density "PD" (Equation (4)) in terms of inhabitants:

$$PD = \frac{\text{number of inhabitants in A}}{\text{Surface area of A } (m^2)} \cdot 100 = \% \text{ population density} \qquad (4)$$

Variable 5. Green Areas: they play an important role in urban sustainability, the development of green economies, adaptation to climate change, and social cohesion [53]. Equation (5) is used to calculate the green area surface ("GA"), where green areas (extracted from the OSM database and complemented with Cartociudad data) are public parks and gardens:

$$GA = \frac{\text{Surface area occupied by green areas in A } (m^2)}{\text{Surface area of A } (m^2)} \cdot 100 = \% \text{ green areas} \qquad (5)$$

Variable 6. Urban Trees: they have historically played an important role in public space, and their relationship with the walkability of a city is evident [54]. The presence of trees in urban areas makes the city more walkable and attractive to citizens and visitors [55]. Defining this variable is performed in two phases: firstly, identifying the presence of vegetation by applying the NDVI index to Sentinel-3 satellite images [56]. Secondly, determine the location of the individual trees using LIDAR data [57], calculating the height and individual crowns of the trees. Urban tree density, which is named "UT", is calculated with Equation (6):

$$UT = \frac{\text{Number of urban trees in A}}{\text{Surface area of A } (m^2)} \cdot 100 = \% \text{ urban trees} \qquad (6)$$

Multi-Criteria Evaluation

Once the six study variables are calculated and homogenized, both in percentage and in raster format, a multi-criteria evaluation is carried out to facilitate the choice of the best alternative in an environment of conflicting criteria and competencies [58]. This evaluation has been completed with the consultation of experts belonging to the following academic and professional fields: (1) nature conservation and environment, (2) college-educated city dwellers, (3) hydrology and climatology, (4) forest engineering, (5) organic chemistry, biochemistry, and environmental chemistry, (6) biodiversity and connectivity. According to the analytic hierarchy process (AHP) [59], each expert (ten for each phase) was asked about the importance of each variable with respect to the others. The AHP process is used due to its robustness and simplicity [60].

The result of the paired comparisons is a square matrix A = ($a_{ij}$), positive and reciprocal ($a_{ij} \times a_{ij} = 1$), whose elements, $a_{ij}$, are an estimate of the proper ratios ($w_i/w_j$) between the priorities associated with the compared elements. The weights of each of the variables are determined using the formula:

$$w_{ij} = \sum_{j=1}^{n} a_{ij} / \sum_{j=1}^{n} \sum_{i=1}^{m} a_{ij} \qquad (7)$$

Further detail about experts' participation can be provided upon reasonable request. With the data obtained from the multi-criteria evaluation, a Saaty matrix [61] is made in order to obtain the weight of each variable in so much per one. The results are: Facilities and services = 0.08; Access = 0.05; Sidewalk width = 0.17; Population density = 0.23; Green areas = 0.29; and Urban trees = 0.18. Finally, with these obtained weights, the final consistency of the ratings is obtained (Table 1). The incompatibility rate in the AHP method is 0.016.

**Table 1.** Saaty Matrix, Phase I experts.

| Expert | FS | AC | SD | PD | GA | UT |
|:------:|:--:|:--:|:--:|:--:|:--:|:--:|
| FS | 1 | 2 | 0.5 | 0.33 | 0.25 | 0.33 |
| AC | 0.5 | 1 | 0.33 | 0.25 | 0.25 | 0.33 |
| SD | 2 | 3 | 1 | 0.33 | 0.33 | 3 |
| PD | 3 | 4 | 3 | 1 | 2 | 0.33 |
| GA | 4 | 4 | 3 | 0.5 | 1 | 4 |
| UT | 3 | 3 | 0.33 | 3 | 0.25 | 1 |

2.2.2. Phase II. Walkability under Climate Pressure 2020

Temperatures in cities are frequently higher downtown than within the surrounding areas because of the effect known as urban heat island [62]. This phenomenon occurs due to the replacement of natural soils by others of an urban nature sealed with dark artificial materials, such as asphalt and concrete [62–64]. In addition, large trees are important factors when assessing microclimatic conditions of an urban environment, in particular regarding surfaces exposed to incident solar radiation and shadows generated [62,65–67]. Therefore, and based on the research of Gluch et al. [65], Ganem et al. [66], Ángel et al. [63] (2010), and Arellano Ramos and Roca Cladera [62], to implement phase II, the following were chosen as climatic variables in the walkability study for Ávila, Salamanca, and Segovia: temperatures, solar radiation, and shadows (Figure 3).

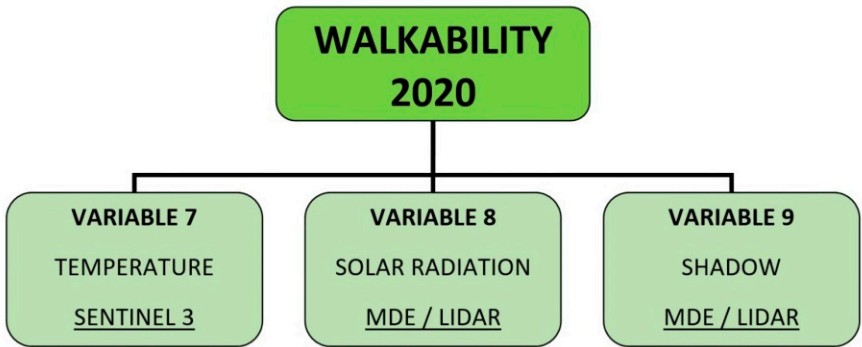

**Figure 3.** Used methodology regarding walkability variables—Phase 2.

*Variable 7. Temperature:* This variable is a function of two sub-variables. The first consists of the mean annual temperatures of the year 2020. Data for this one is obtained from the Agroclimatic Atlas of Castilla y León. The other one consists of the calculation of the median temperatures of the last 5 years, from 2016 to 2020, which were decided to be used since they are not affected by extreme values [68].

*Variable 8. Solar Radiation:* calculated according to the Spain annual insolation map of Viedma Muñoz and Capel Molina [69]; the Duero River basin is in an area of high solar radiation with more than 2600 useful hours. The maximum daily irradiation values are reached in June and July, with more than 20 MJ/m$^2$. The minimums are concentrated in December and January, with values below 10 MJ/m$^2$ [70]. In a small geographical area with similar climatic conditions, it can be accepted that the differences in insolation between two points on the same day are exclusively conditioned by the relief, so that can be approached from the digital elevation model [71].

*Variable 9. Shadows:* To develop the analysis of this variable, a digital elevation model (DEM) was used to create a layer of shadows of buildings, trees, and other urban elements. Subsequently, a hillshade layer has been created, also through the digital elevation model, which shows the shading of slopes on the surface of the territory. Finally, Equation (7) is applied:

$$Shadows\ variable = \frac{Hillshade + Shadows}{2} \tag{8}$$

Multi-Criteria Evaluation

Once the three climate-related study variables are computed and homogenized, a multi-criteria evaluation is performed to facilitate the choice of the best alternative in an environment of conflicting criteria and competencies. The multi-criteria evaluation (Table 2) is performed with eight experts analogously to Phase I.

**Table 2.** Saaty matrix, Phase II experts.

| Expert 1 | T | RS | S | Expert 2 | T | RS | S |
|---|---|---|---|---|---|---|---|
| T | 1 | 0.33333 | 0.33333 | T | 1 | 0.25 | 0.25 |
| RS | 3 | 1 | 0.5 | RS | 4 | 1 | 0.5 |
| S | 3 | 2 | 1 | S | 4 | 2 | 1 |
| Expert 3 | T | RS | S | Expert 4 | T | RS | S |
| T | 1 | 0.25 | 0.33333 | T | 1 | 0.33333 | 3 |
| RS | 4 | 1 | 0.5 | RS | 3 | 1 | 3 |
| S | 3 | 2 | 1 | S | 0.33333 | 0.33333 | 1 |
| Expert 5 | T | RS | S | Expert 6 | T | RS | S |
| T | 1 | 0.5 | 0.5 | T | 1 | 0.33333 | 0.5 |
| RS | 2 | 1 | 0.5 | RS | 3 | 1 | 0.5 |
| S | 2 | 2 | 1 | S | 2 | 2 | 1 |
| Expert 7 | T | RS | S | Expert 8 | T | RS | S |
| T | 1 | 3 | 3 | T | 1 | 0.25 | 0.2 |
| RS | 0.33333 | 1 | 0.5 | RS | 4 | 1 | 0.5 |
| S | 0.33333 | 2 | 1 | S | 5 | 2 | 1 |
| Average | T | RS | S | | | | |
| T | 1 | 0.33333 | 0.25 | | | | |
| RS | 3 | 1 | 0.5 | | | | |
| S | 4 | 2 | 1 | | | | |

Thus, the weights obtained for the three Phase II variables are the following: temperatures = 0.12, solar radiation = 0.34, and shadows = 0.54.

The weighted linear sum enables the GIS tools to aggregate all the criteria quickly and easily. By using a weighted linear sum and the determined values from the experts (Table 2), the data from the temperature, solar radiation, and shadows variables were combined. The information is compiled in the map results, which also indicate how walkable the city is. The following equation was used to calculate the walkability index for the three studied cities:

$$\text{Walkabiilty } 2020 = \alpha \times \text{shadows} - \beta \times \text{solar radiation} - \gamma \times \text{temperature } 2020. \quad (9)$$

which "$\alpha$" takes the average value given by the experts to the shadows (0.54), "$\beta$" will be the average value given to the solar radiation (0.34), and "$\gamma$" will be the average value for the temperature (0.12).

2.2.3. Phase III. Walkability under Future Climate Pressure 2020

In this phase, several projections of different future climate change scenarios are performed to compare them with the current scenario. Thus, to develop the projections for the 21st century, the following scenarios were selected from the emissions scenarios provided by the AEMET in the guide to regionalized climate change scenarios for Spain based on the results of the IPCC-AR4 (2014): A1B and A2. Scenario A1B is the consequence of a situation of moderate emissions and rapid economic growth based on the balanced use of conventional and renewable energies. Scenario A2 is the consequence of a situation of high emissions resulting from the massive use of fossil resources and slow implementation of renewable energies. In this paper, in order to get models closer to the regional reality, current climate data from Nafría García et al. [72] was used as a basis. The maps for the mean annual temperature variable are obtained for the current situation and for the MIROC-H model projections, which have been made in the two scenarios (A1B and A2) for the years 2030, 2050, and 2100.

In this way, the current walkability value is updated with the projected temperature for each climatic scenario considered in this study, following Equation (10).

$$\text{Walkability scenario x} = \alpha \times \text{shadows} - \beta \times \text{solar radiation} - \gamma \times \text{temperature scenario x.} \tag{10}$$

where x = the climatic scenario considered (scenarios A1B and A2 for the years 2030, 2050, and 2100).

## 3. Results

This section shows the results obtained for each phase of the methodology, obtaining walkability values based on the defined walkability index during Phase 1 for the three cities, while in Phases 2 and 3, the results of the index are modified according to the climate-related variables. Higher values indicate better walkability.

### 3.1. Phase 1. Walkability

Once the modeling was carried out with the six previously mentioned variables, the weighted values obtained through expert consultations were applied. Figure 4 shows the results. The maps obtained for the three cities show very high walkability values in recently built residential neighborhoods and downtown areas.

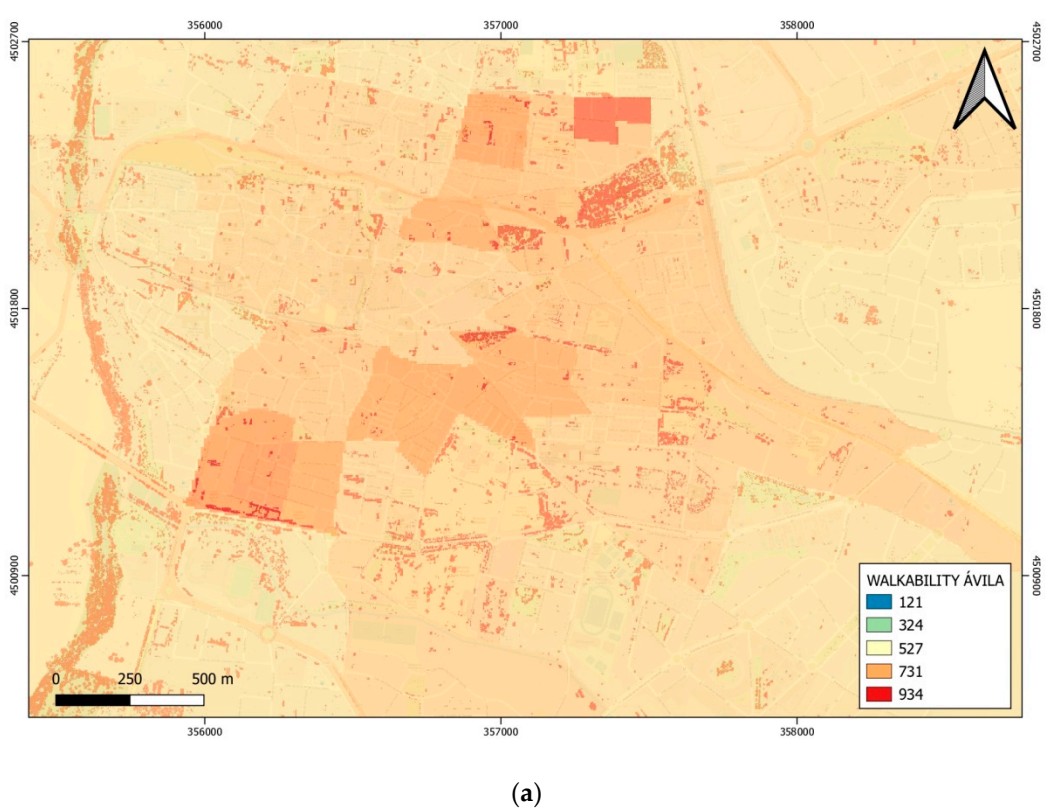

(**a**)

**Figure 4.** *Cont.*

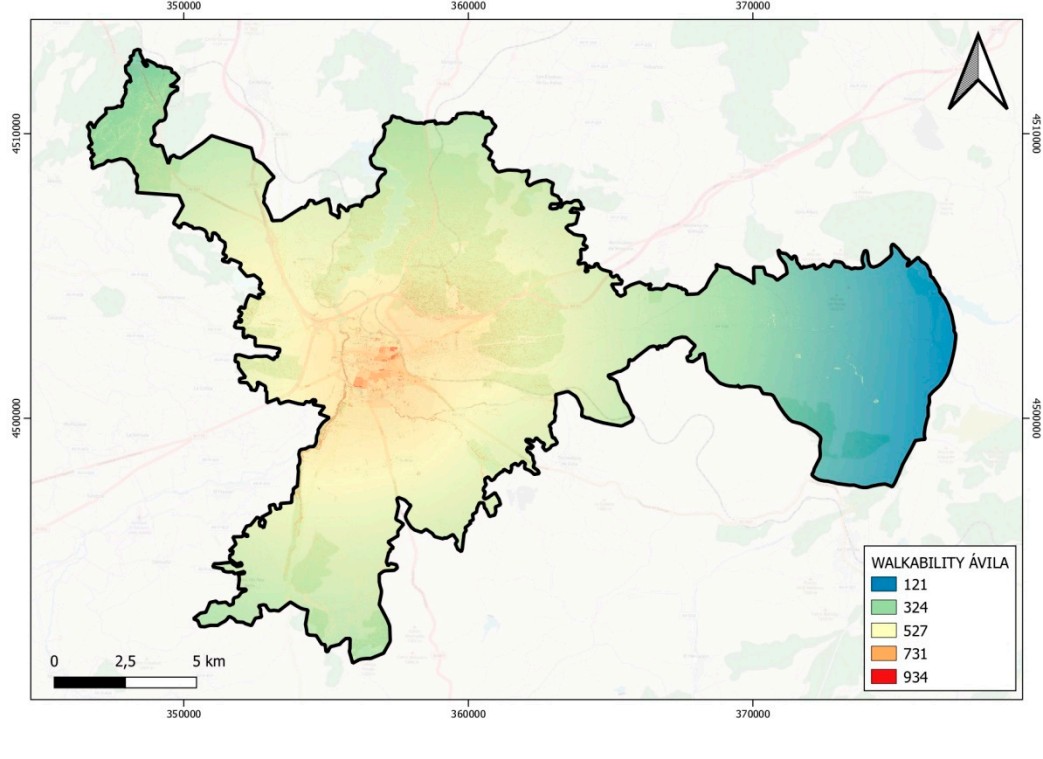

(**b**)

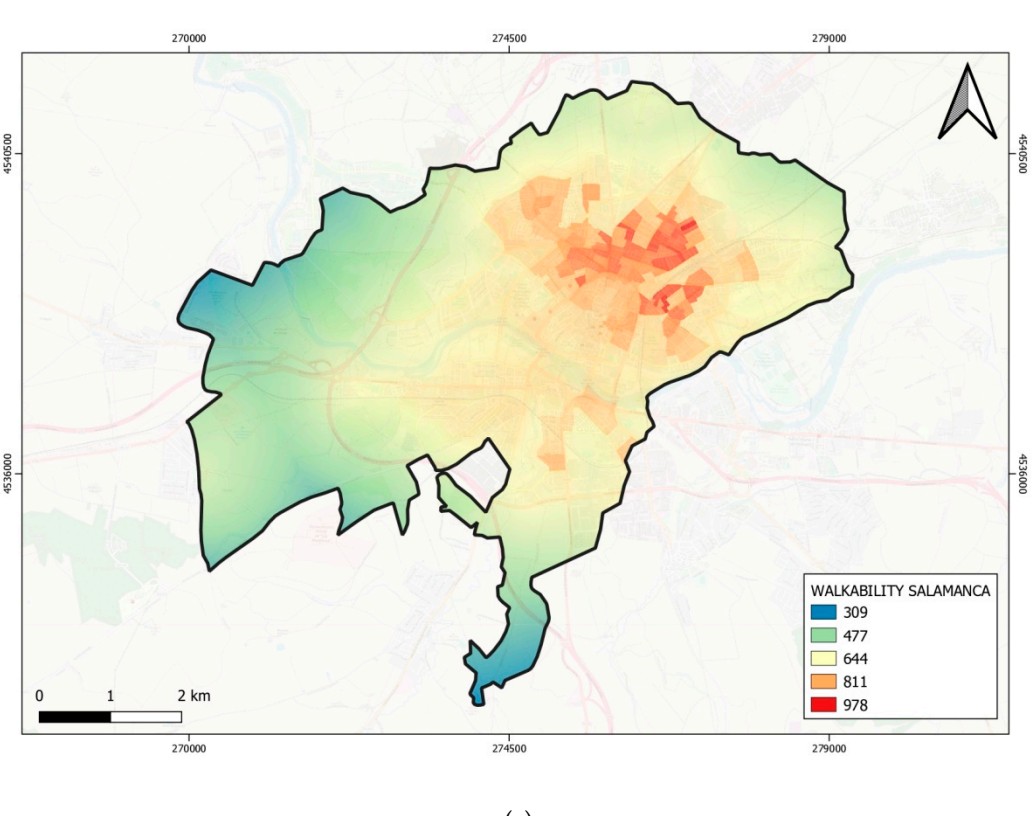

(**c**)

**Figure 4.** *Cont.*



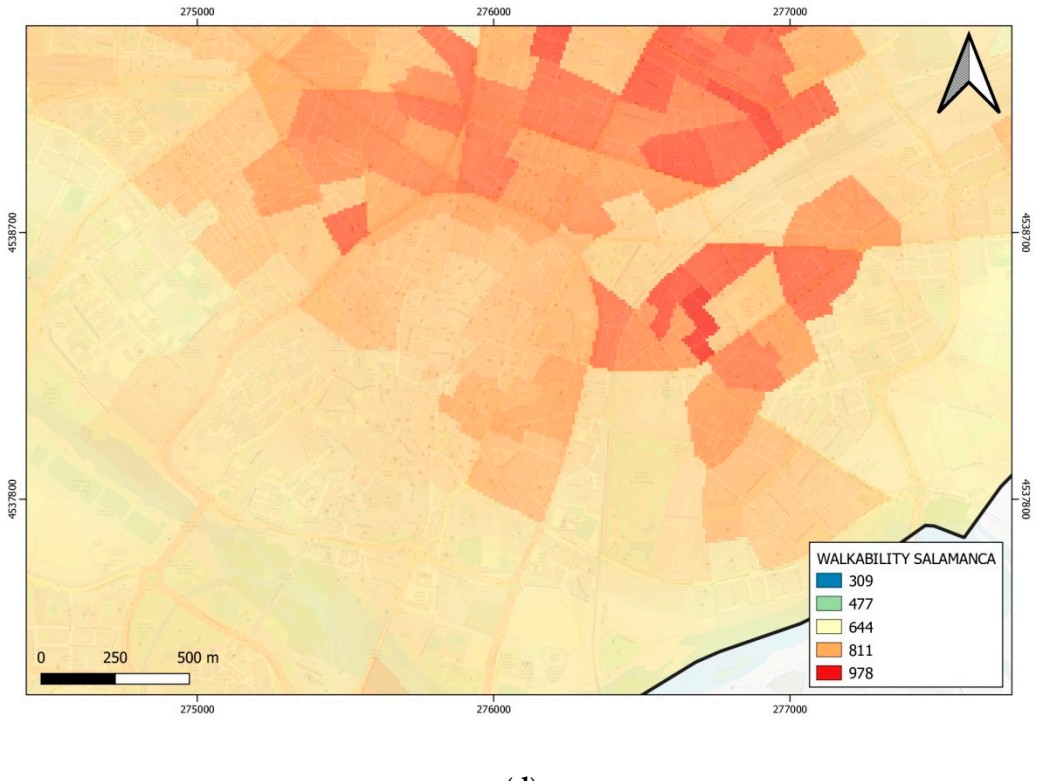

(**d**)

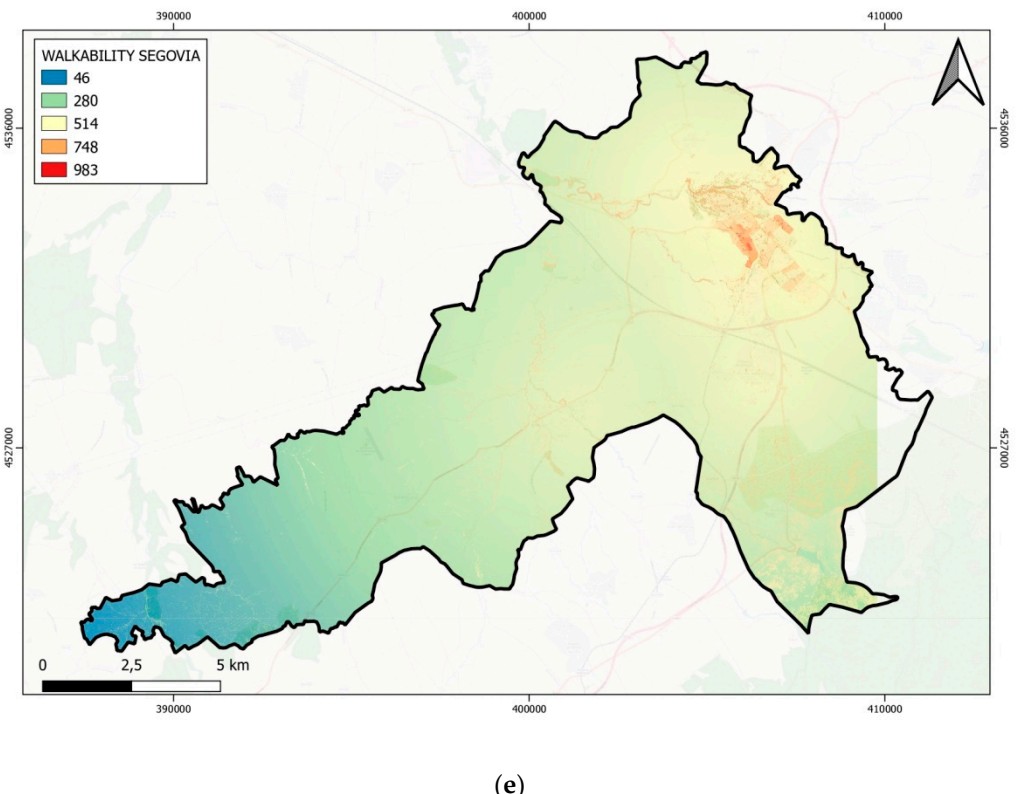

(**e**)

**Figure 4.** *Cont.*

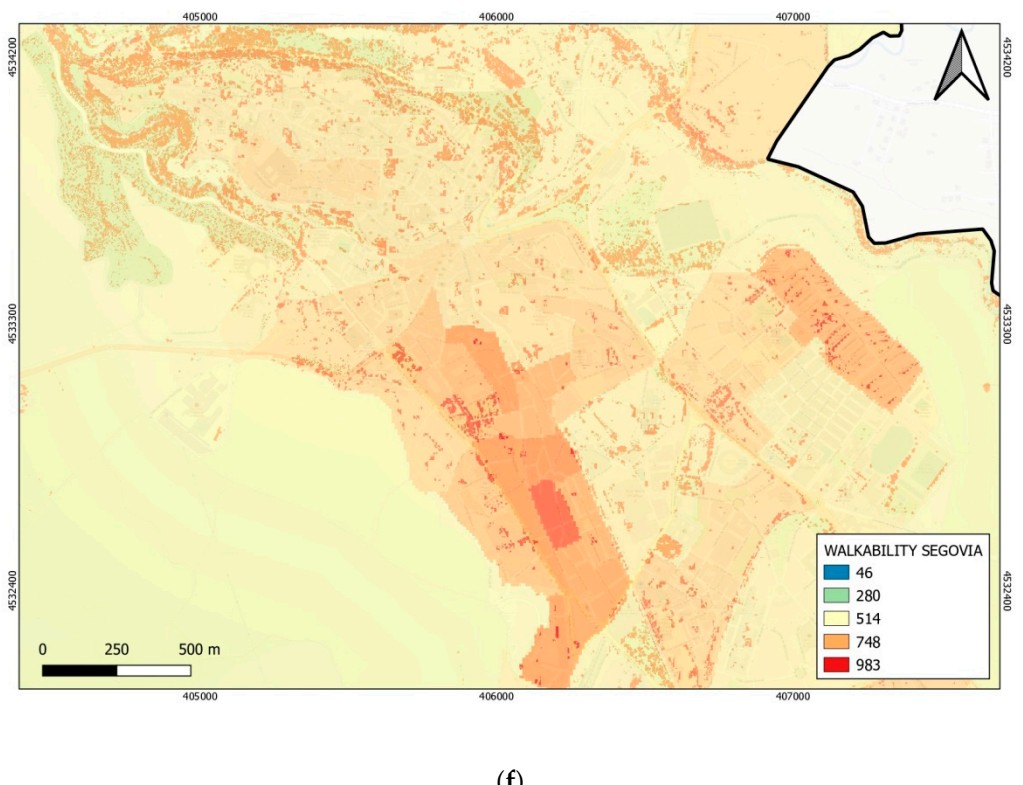

(**f**)

**Figure 4.** (**a**) Walkability in the city of Ávila, (**b**) Walkability in Ávila (1:12,000), (**c**) Walkability in the city of Salamanca, (**d**) Walkability in Salamanca (1:12,000), (**e**) Walkability in the city of Segovia, (**f**) Walkability in Segovia (1:12,000).

*Variable 1. Facilities and Services:* the modeling performed considering the previous assumptions show an excellent walkability index in the historical centers of the three cities and in some of the residential neighborhoods of Salamanca. This variable can be considered optimal for the inhabitants of the areas indicated and for the influx of tourism. However, the results for the annexed neighborhoods (localities that were previously independent villages and now depend on the city council for their management) of Segovia and Ávila, as well as for the periphery of Salamanca, are not favorable.

*Variable 2. Accessibility:* a good index of walkability was obtained in the three downtowns, areas of greatest tourist affluence, although there are certain disparities in the three cities. The worse values were obtained in the peripheries, especially in the cases of Ávila and Segovia. This is mainly due to the pedestrianization of the historic centers in Ávila, Segovia, and Salamanca in recent decades. This process has been implemented in most of the World Heritage cities since traffic on streets with deficient widths for the simultaneous transit of pedestrians and vehicles, as well as their parking, causes worse walkability values.

*Variable 3. Sidewalk:* the maximum values of walkability offered by this variable are highest in the old town and in the recently built neighborhoods and gradually decrease as the distance to these areas increases, reaching minimum values in other peripheral neighborhoods. The city center offers walkable sidewalks in non-pedestrian areas, but in general, these are of lesser width due to the narrowness of the streets, whose layout responds to medieval and even ancient Roman urban planning criteria. Neighborhoods built during the 1990s and the first decade of the twenty-first century, on the other hand, offer wide sidewalks that allow excellent mobility.

*Variable 4. Population Density:* the model showed that the highest population densities in the three cities are in the downtown areas, followed by the newly built neighborhoods.

At the other extreme are the industrial and annexed neighborhoods in the cases of Ávila and Segovia, where population densities are very low.

*Variable 5. Green Areas:* results showed an excellent score in newly built residential neighborhoods, whose planning includes significant areas dedicated to green spaces. The downtown areas, which are the most popular for tourism, also obtained high scores. Industrial areas and annexed neighborhoods, on the other hand, showed very low values.

*Variable 6. Urban Trees:* trees between 4 and 30 m in height were included, and the maximum score was given to the presence of trees and proximity to each other. As a result, the Adaja and Eresma riverbanks in Ávila and Segovia, respectively, stand out as the most walkable areas in relation to this variable.

The Overall Result of Phase I Variables

The results obtained for walkability place Salamanca as the best city of the three studied. It corresponds to the city with the highest population density.

*3.2. Phase 2. Walkability under Climate Pressure 2020*

*Variable 7. Temperatures:* data for June in the period 2016–2020 was collected from Sentinel 3. When analyzing the data for these five years and comparing them with the June 2020 temperature, a slight increase was observed, relatively generalized in the three municipalities.

*Variable 8. Solar radiation:* The solar radiation data were collected on 21 June 2020 between 14:00 and 15:00 (UTC+1), the first zenith after the summer solstice and, therefore, the time of greatest insolation of the year. The analysis and modeling of these data have provided a maximum solar radiation map for the municipalities of the three cities (Figure 5). Thus, any other scenario will have lower insolation, making it possible to evaluate the effect of climate on walkability.

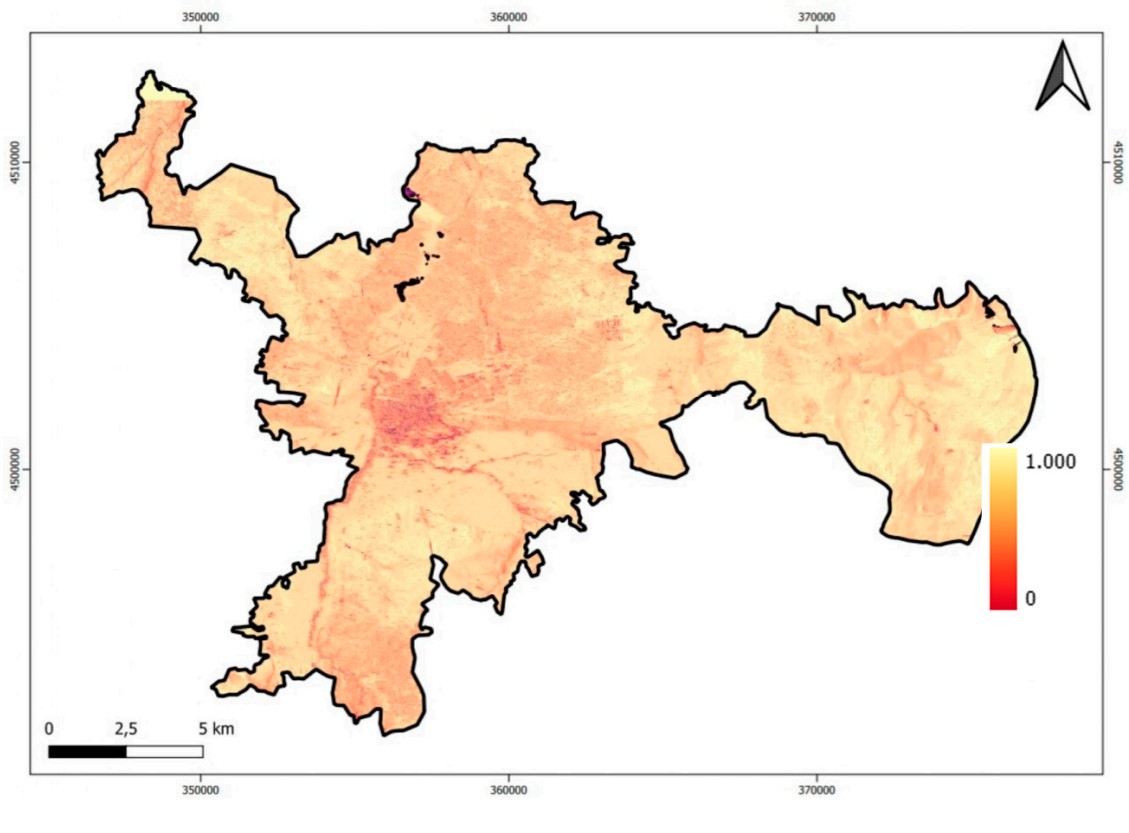

(**a**)

**Figure 5.** *Cont.*

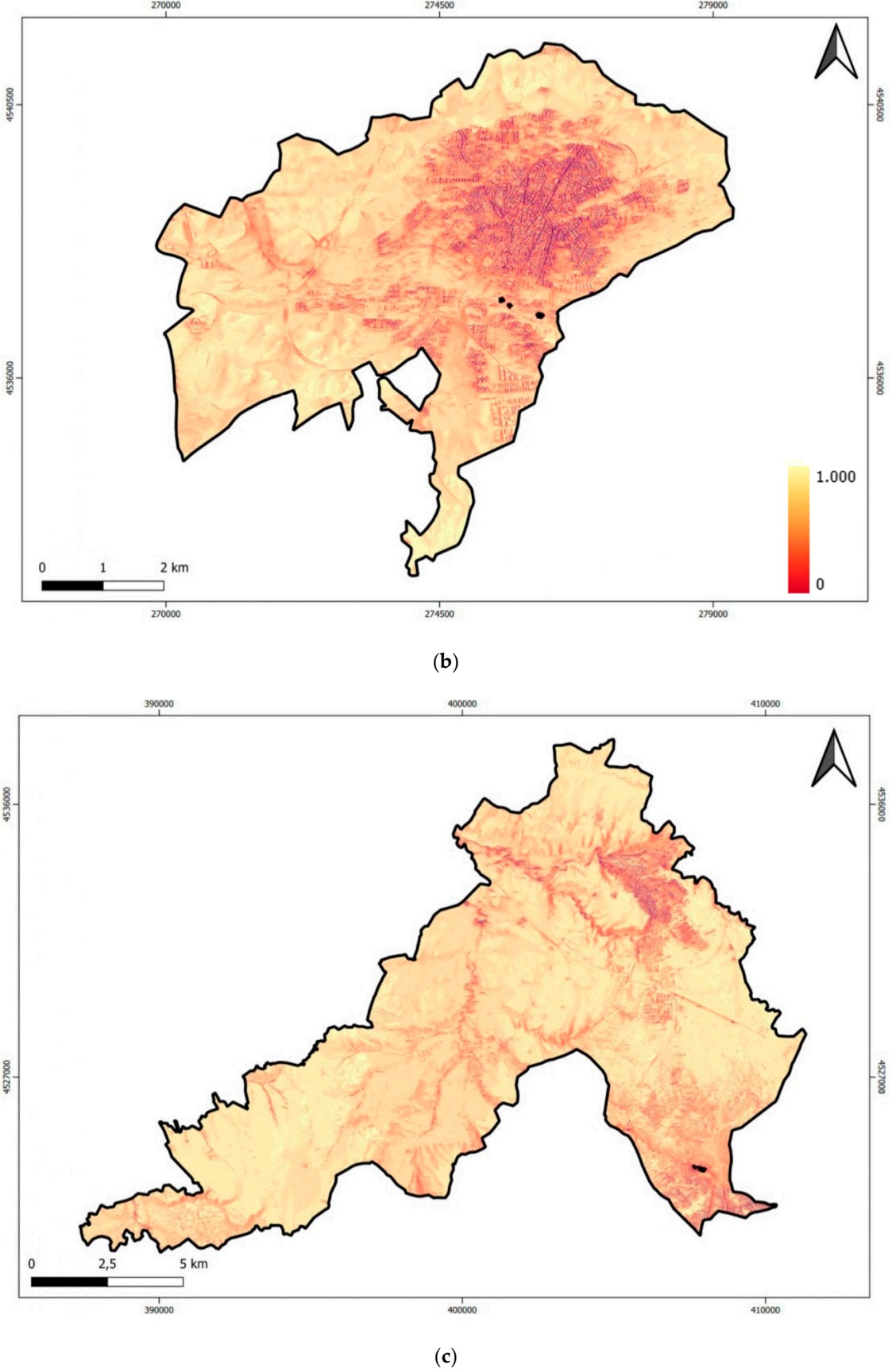

(**b**)

(**c**)

**Figure 5.** (**a**) Solar radiation in the city of Ávila, (**b**) Solar radiation in the city of Salamanca, (**c**) Solar radiation in the city of Segovia.

*Variable 9. Shadows:* the obtained results show three areas where the covered shadow surface is especially significant. Firstly, central areas stand out where the streets are narrow, and there is a greater agglomeration of buildings. Secondly, the index of shaded areas is also high in green areas since trees intercept solar radiation. Figure 6 shows the results for this variable.

The Overall Result of Phases I and II Variables

Once the modeling of the variables included in Phases I and II was accomplished, the result obtained is presented in Figure 6, which shows maps of walkability under climate pressure in 2020. The most walkable areas correspond, again, to the downtowns of older cities, where many areas reach values close to 1000, which is very positive for tourism, the main source of income for World Heritage cities [73].

The peripheral areas and annexed neighborhoods obtained very low walkability values, like in Phase I, due to the scarcity of shadows that make them very unwalkable on the hottest days. In conclusion, Phase II shows similar results to Phase I. However, one difference is worth noting. Recently built neighborhoods, which had very good walkability values in Phase I, have been penalized in Phase II because of the large width of the streets and dispersion of buildings, which results in a reduction in shaded areas and an increase in the incident solar radiation.

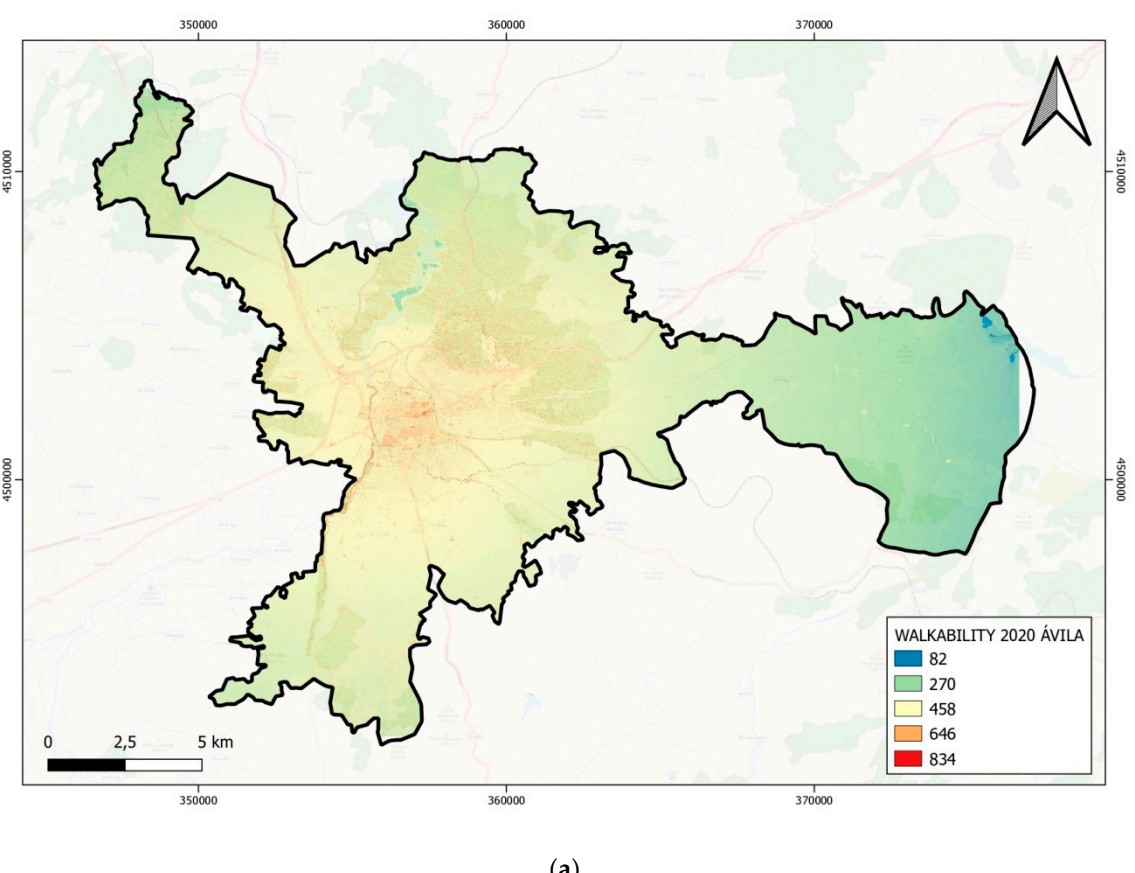

(**a**)

**Figure 6.** *Cont.*

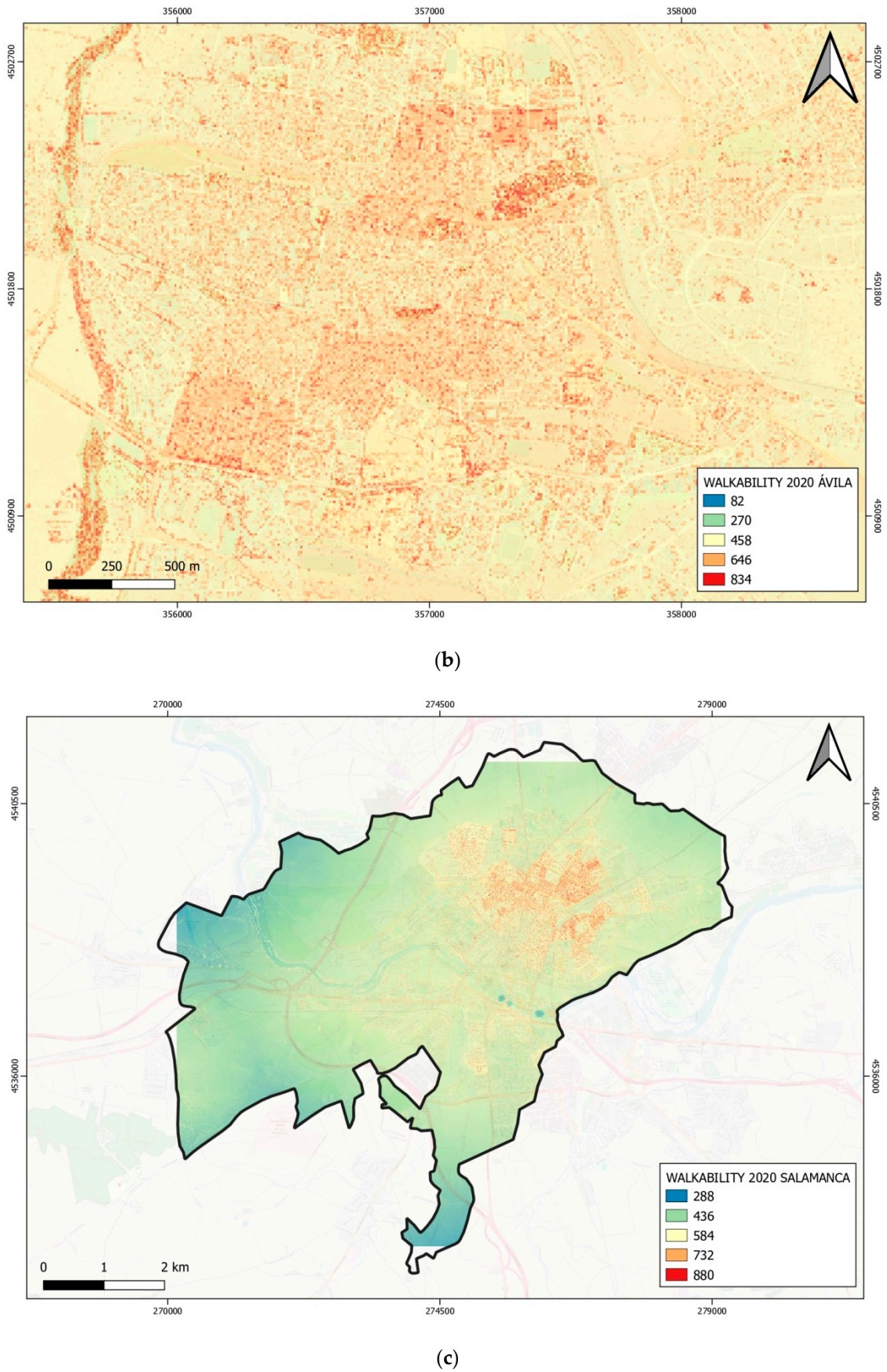

(**b**)

(**c**)

**Figure 6.** *Cont.*

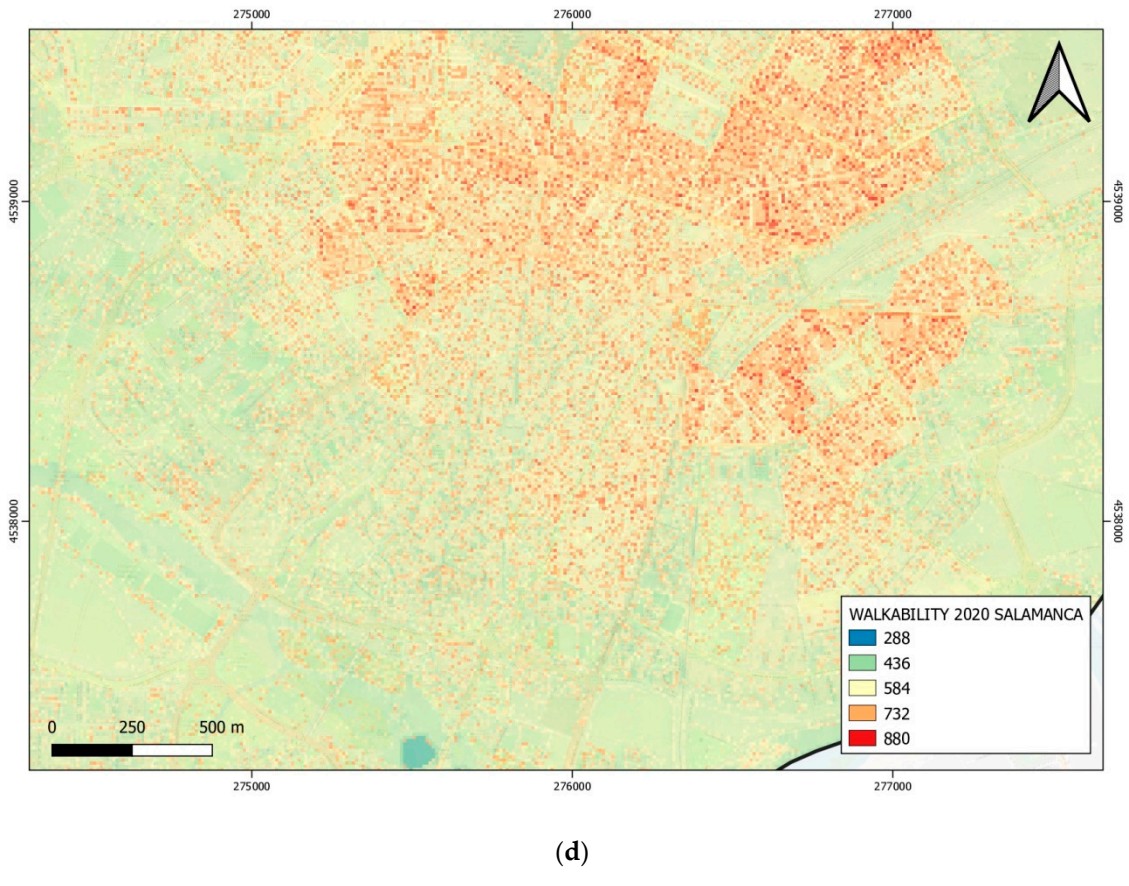

(**d**)

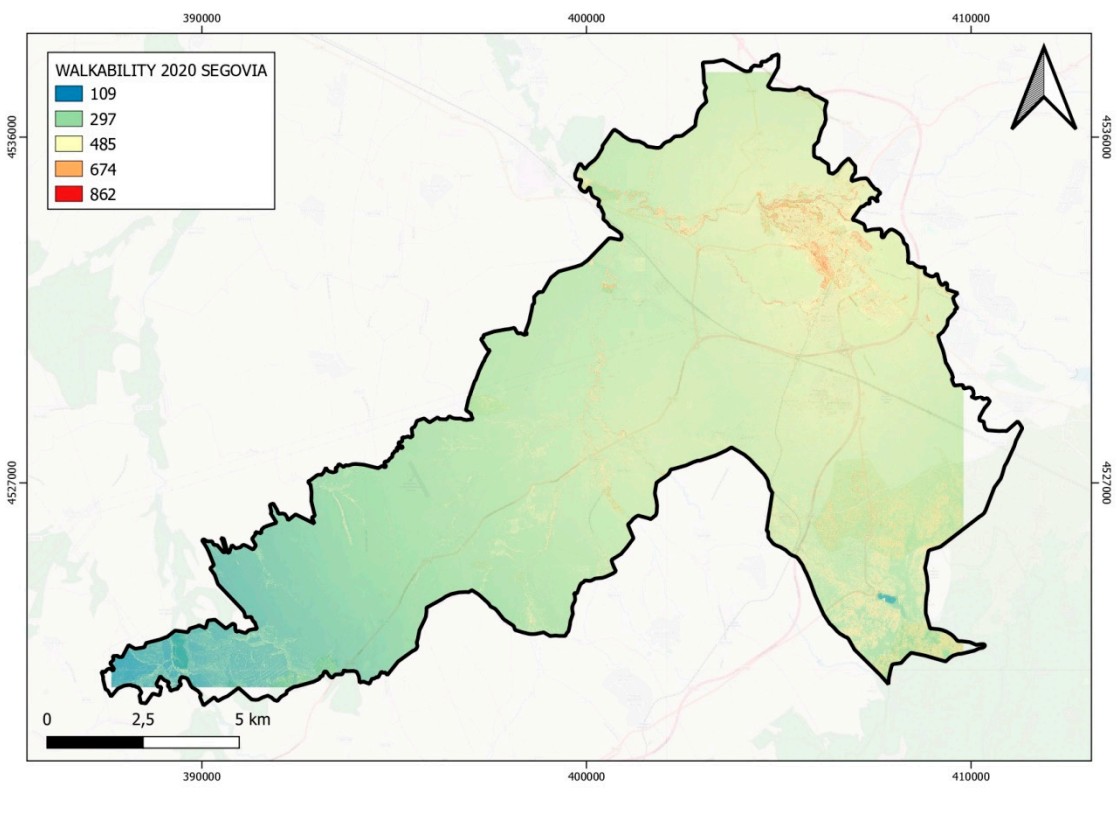

(**e**)

**Figure 6.** *Cont*.

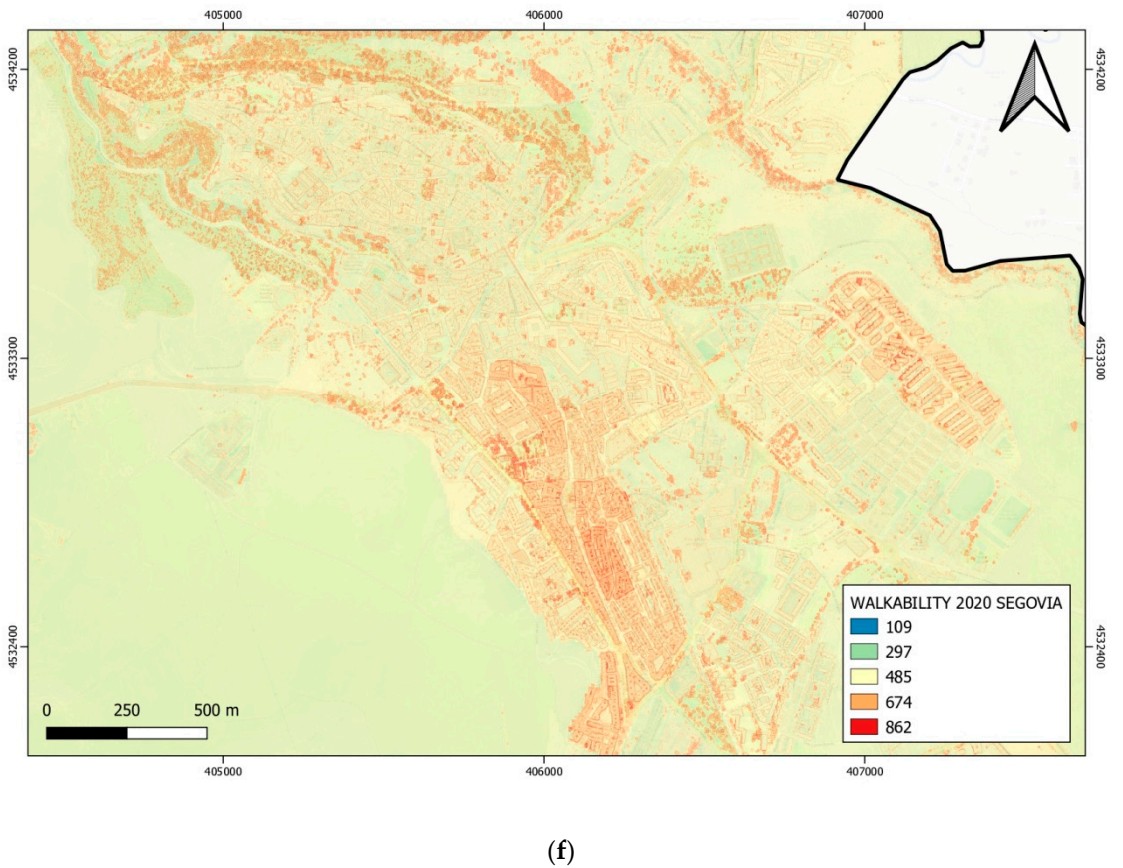

(**f**)

**Figure 6.** (**a**) Walkability 2020 in the city of Ávila, (**b**) Walkability 2020 in Ávila (1:12,000), (**c**) Walkability 2020 in the city of Salamanca, (**d**) Walkability 2020 in Salamanca (1:12,000), (**e**) Walkability 2020 in the city of Segovia, (**f**) Walkability in 2020 Segovia (1:12,000).

*3.3. Phase 3. Walkability under Climate Pressure in Future Scenarios*

Figure 8 and Appendix A shows the walkability results under future climate pressure for Ávila. It includes six possible scenarios: scenarios A1B (moderate emissions) and A2 (high emissions) for 2030; scenarios A1B and A2 for 2050; and scenarios A1B and A2 for 2100. Although any future projection is subject to some uncertainty, there is a generalized decrease in the walkability index in the three municipalities regarding the 2020 scenario. On the other hand, tourist areas remain the most walkable areas, followed by newly built neighborhoods, although the latter are more penalized by the increase in temperatures due to the scarcity of shadows. Figure 8 shows differences in the walkability more significant as time passes because the temperature increases faster in the high emissions scenario. This modeling shows another consequence of climate change, the reduction in the walkability index of three World Heritage cities, which can be extrapolated to other historic cities with similar characteristics.

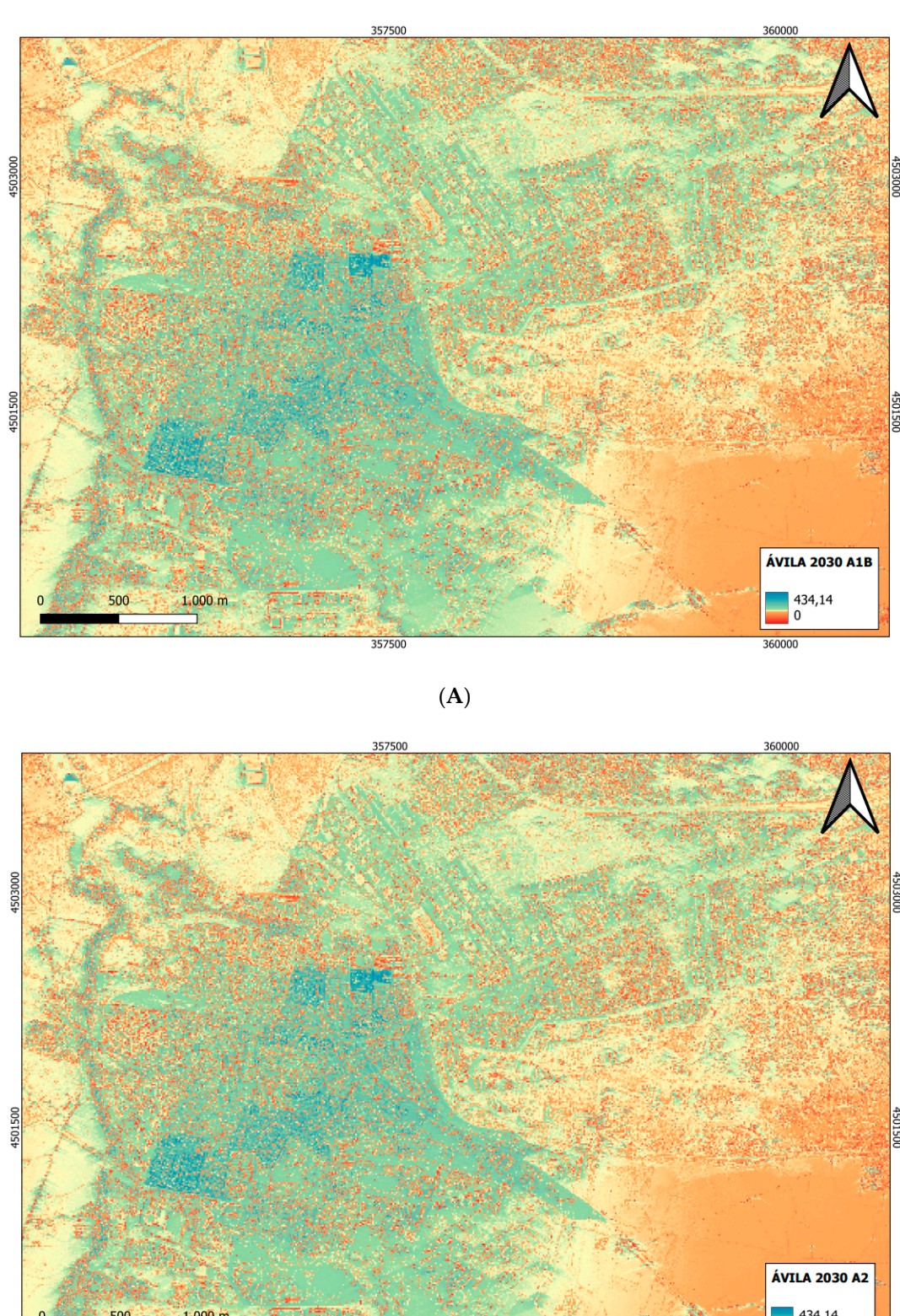

(**A**)

(**B**)

**Figure 7.** *Cont.*

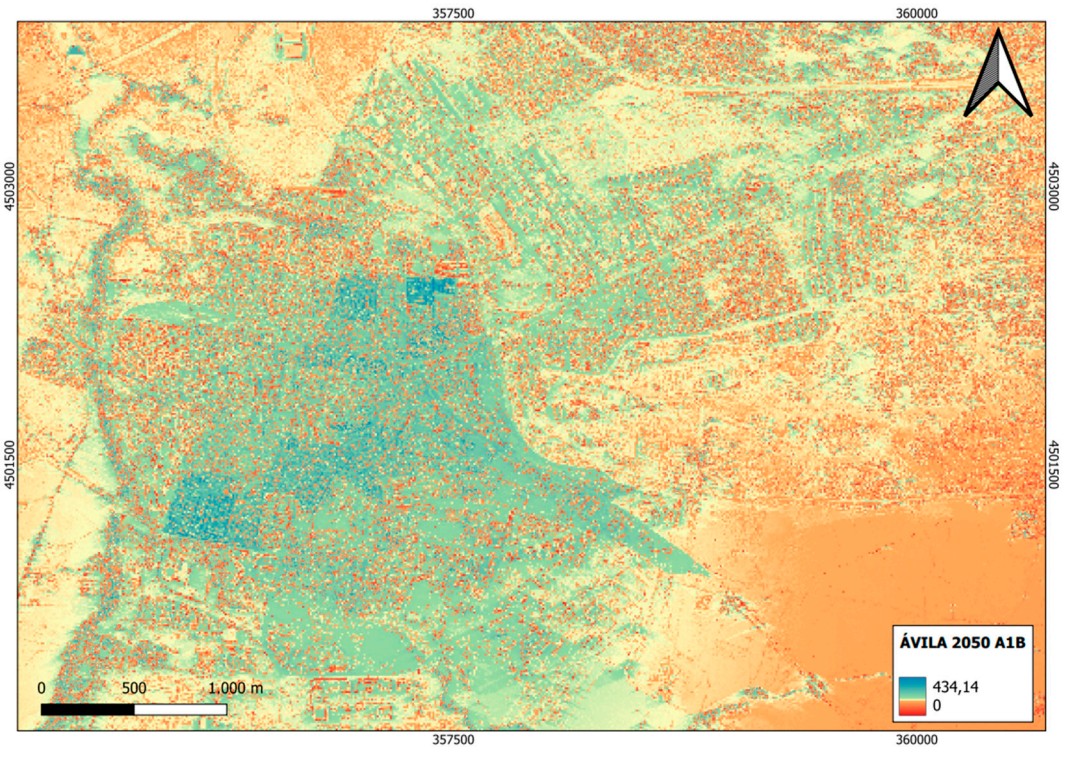

(**C**)

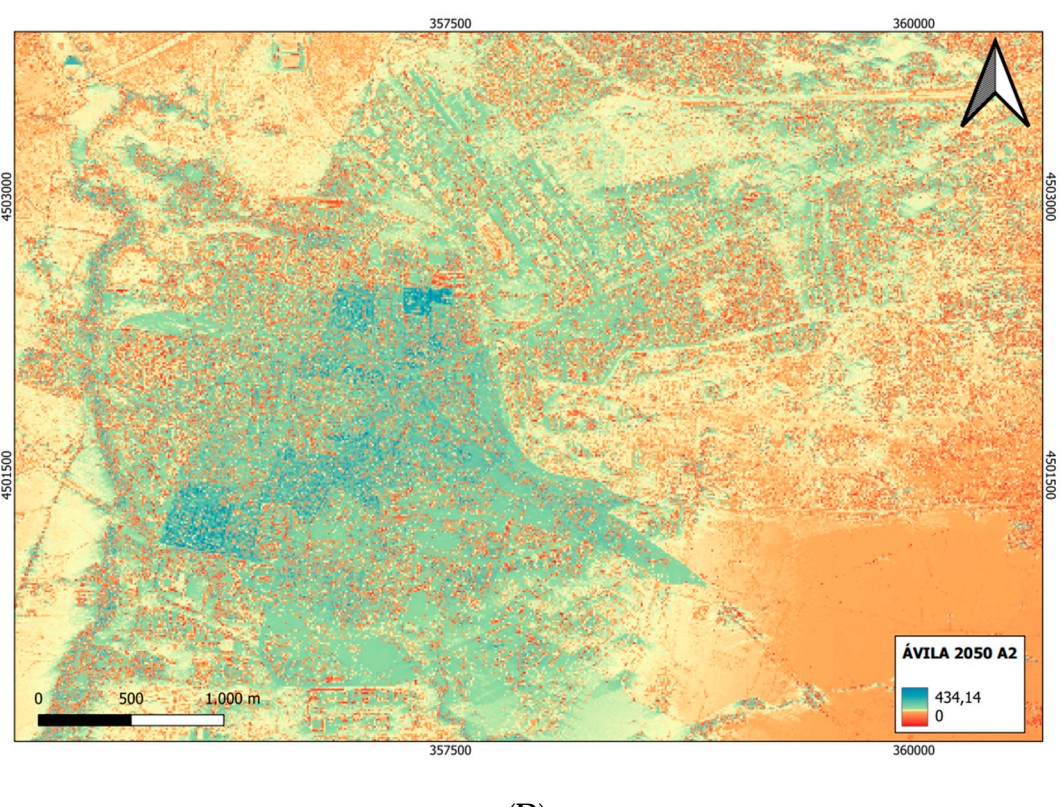

(**D**)

**Figure 8.** *Cont.*

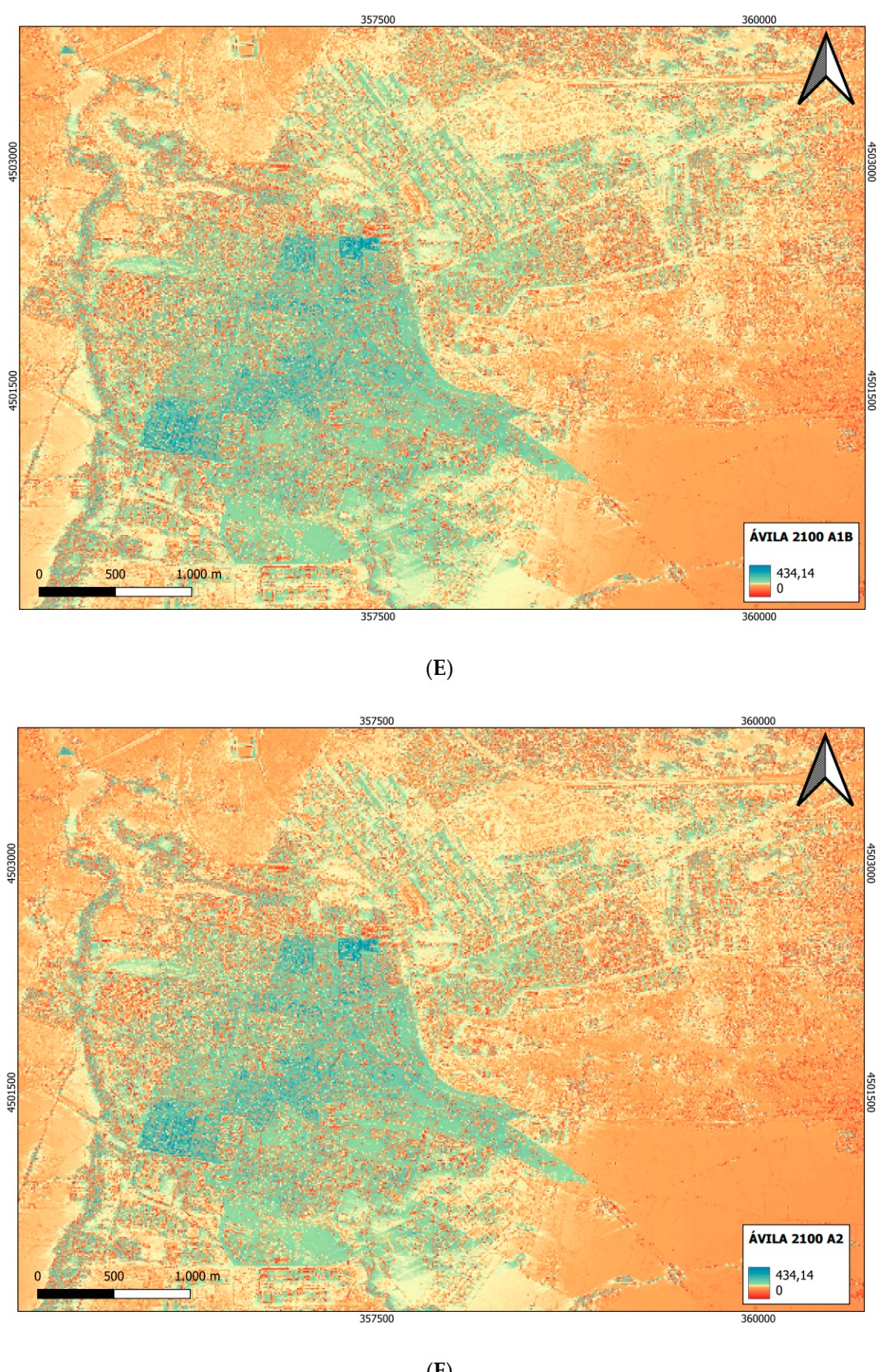

(**E**)

(**F**)

**Figure 8.** (**A**) Walkability 2030 A1B in the city of Ávila, (**B**) Walkability 2030 A2 in Ávila, (**C**) Walkability 2050 A1B in Ávila, (**D**) Walkability 2050 A2 in Ávila, (**E**) Walkability 2100 A1B in Ávila, (**F**) Walkability 2100 A2 in Ávila.



## 4. Discussion

The historical centers of the three studied Castilian cities have their origins in the Ancient Ages. However, despite the obsolescence of the building and the very old urban planning, these traditional neighborhoods currently present a walkability model that responds to the new demands of current urbanism [74]. Higher walkability indices have been correlated with better tourism within cities [32]. This is excellent news in terms of tourism management since the areas that receive the most visitors are precisely those that are the most walkable.

In contrast to the central areas and new neighborhoods, the peripheral areas with buildings constructed in the second half of the twentieth century, industrial estates, and annexed neighborhoods have very low walkability values, sometimes up to 90% lower than those offered by the central areas. Planning at the municipal level, therefore, must address these deficiencies and try to alleviate them as far as possible. Blečić et al. [75], for example, suggest that the concept of walkability can be used as an indicator of peripherality. Finally, it should be noted that in the case of the cities under study, the optimal planning of new neighborhoods, built far from the center since the 1990s but with very good walkability values, should be appreciated. These would constitute an exception to the proposal of Blečić et al. [75].

The fact that tourist areas have an optimal walkability index gives these cities a potential that they should undoubtedly take advantage of. Moreover, it is appropriate for them to promote this resource since most of the research on walkability focuses on the effects of walkability on the social, health, economic, and environmental spheres, but few are interested in the effects on tourism [76].

Some aspects of the variables chosen and considered for this study will now be discussed:

Phase I. Walkability: **Variable 1. Facilities and Services:** the proximity between different facilities and services is an important factor when calculating walkability [3]. Public transport also has a strong influence on walkability, as it enables the population to access services [77]. **Variable 2. Accessibility:** the distance between homes and establishments is an important factor for walkability [30,52]. For this reason, they have been considered when performing this study, and the proximity to footpaths, pedestrian streets, paths, and stairs have been positively scored. **Variable 3. Sidewalk**: Sidewalk width and distance to them are other variables that influence walkability [47,78–80]. **Variable 4. Population Density**: Population density is another factor to consider when calculating walkability [3,47,54]. None of the three cities has an excessively high population density, but effectively, the highest walkability values correspond to the most densely populated areas. Predictions on the evolution of population density make it possible to anticipate the needs of public services, such as the health or education system, as well as forecast future demand for employment and productivity [81]. **Variable 5. Green Areas:** The presence of green areas provides a series of benefits, among others, the capture of carbon dioxide, contributing to the mitigation of climate change. There are several papers on walkability indices that regard it as a factor to be considered [6,47,82,83]. **Variable 6. Urban Trees:** the presence of trees in cities provides several ecosystem benefits, such as the capture of carbon dioxide and other gases harmful to the atmosphere [84].

Furthermore, the climatic variable has been considered one of the most relevant variables for calculating walkability indices [32]. In Spain, an analysis of recent temperature trends confirms that there has been a generalized increase in the mean annual temperature since the mid-1970s, mainly in winter, slightly higher than that observed globally [70]. This fact justifies Phase II, and some points of the chosen variables have been discussed. **Variable 7. Temperatures:** they were included in the walkability index because, among the multiple consequences of climate change, one of the most relevant is the global increase in average temperature [85]. **Variable 8. Solar Radiation:** another consequence of climate change is the variation in atmospheric composition and dynamics that leads to a modification in the amount of solar radiation reaching the surface of the Earth, whose increase leads to anticyclonic situations with less cloudiness, less precipitation, lower minimum

temperatures, and higher maximum temperatures [86–88]. **Variable 9. Shadows:** The increase in temperatures that climate change implies can be a problem when going for a walk, especially on warmer days. For this reason, it is necessary to include an analysis of the shadows in the calculation of the walkability index [6,89–91]. In this way, future studies should consider the limitation in assuming walkability to be linearly decreasing in temperature because the projected warming would improve walkability in winter but worsen it more drastically during summer heatwaves, too.

Climate change is modifying the way that people relate to the environment, and weather patterns considered relatively stable over the past centuries are changing rapidly [92]. For this reason, city planning must consider possible future climate scenarios, and they are in Phase 3. Although each studied city has its own characteristics, all three adapt relatively well to the sprawl city model, as opposed to the compact city model, which is much more efficient and less polluting [93]. The sprawl city model is characterized by low-density development, with housing, shopping centers, and other amenities spread out over a wide area. This type of development tends to rely heavily on automobiles and can lead to long commute times, traffic congestion, and other negative impacts on the environment and quality of life. The compact city model, on the other hand, is characterized by higher-density development, with buildings and amenities located closer together. This type of development is generally seen as more sustainable and efficient, as it can reduce reliance on automobiles and promote public transportation and walkability.

The implication is that the sprawl city model is less effective and more damaging than the compact city model. The dispersed city model presents low density with no continuity among its parts, the different uses are distant, and the periphery, occupied mainly by single-family homes, lacks basic services and facilities [94].

In addition limitation in assuming walkability to be linearly decreasing in temperature—the expectation would be that warming would improve walkability in winter but worsen it more drastically during summer heatwaves.

## 5. Conclusions

This study determined the walkability in three World Heritage cities in central Spain: Ávila, Salamanca, and Segovia. The methodology was developed in three phases. The first phase focused on the calculation of a current index of walkability according to the proposal of Rattan et al. [48], and for each of the cities were analyzed the following variables: facilities and services, access, sidewalk width, population density, green areas, and urban trees. In the second phase, a new walkability index was calculated for the three cities, with the previous variables and the following climate-related variables: temperatures, solar radiation, and shadows. Finally, in the third phase, predictions were made of the climate scenarios in the years 2030, 2050, and 2100 and these predictions were compared with the climate scenario obtained in Phase II. In conclusion, this paper presents a new methodology that interrelates three elements: walkability, climate change, and tourism. It also allows the elaboration of predictions of future scenarios to maintain and even improve the walkability index in urban cities under climate pressure.

In the three cities, the areas with the highest walkability correspond to the historic center of the cities, while the peripheral areas have a generally lower index. The best conditions in terms of the variables analyzed coincide with central areas, with better accessibility to facilities, higher population density, and a greater number of pedestrian streets. In the Walkability 2020 calculation, less insolation is observed in vegetation cover areas corresponding to the central areas due to trees and buildings located there. This shows the path for mitigation measures for climate change to enhance the tourism impact through the use of new or improved vegetation-covered routes that would be considered in further research.

An increase in temperature compared to the last 5 years suggests an increase in solar radiation in cities coherent with climate change. Expert consultations carried out for the nine variables ensure good consistency, with values close to 0.1 in all cases. The study and the comparison of the different scenarios for 2030, 2050, and 2100 show a progressive

increase in temperatures due to climate change, which should be the focus of the future in the infrastructure and design of cities. This analysis raises awareness of the problem of climate change and the need to adopt appropriate measures to mitigate the effects.

Even when the main conclusions could seem like common sense, it is especially valuable the information provided by the walkability index about the effect of climate variables shows the path for ulterior studies to replicate the index in other cities and climate conditions. Nevertheless, the authors acknowledge that other variables, such as noise or pollution levels, could modify the walkability values, but those data were not available at the time of this research.

## 6. Study Limitations and Prospective Research Lines

Among the limitations of this research, it was possible to highlight the difficulty of straightforward connecting—considering the lack of a powerful indicator—that links walkability and tourism, which is an opportunity for future researchers to invert this link definition. Moreover, as is the case, studying more references to tourism—i.e., the size of the facilities and the number of tourists—are just a few examples.

The same goes for additional climate data that, if added, could give us a complete picture of the climate of the sites under research—i.e., the short time series given for the air temperature. In fact, to calculate average temperature 10-year series of annual and monthly mean temperatures, which in this study was seen as some limitation and potential future research opportunity to advance in the thematic scientific field.

**Author Contributions:** Conceptualization, J.V. and D.G.; methodology, I.G., J.V., D.G. and V.R.; validation, A.H., R.A.C. and F.H.; formal analysis, J.I., J.V. and V.R.; investigation, V.R., J.V. and D.G.; data curation, A.H. and F.H.; writing—original draft preparation, J.I., I.G. and J.V.; writing—review and editing, V.R., J.V., D.G. and R.A.C.; visualization, J.I. and D.G.; supervision, J.V., A.H., F.H. and R.A.C. project administration, J.V. All authors have read and agreed to the published version of the manuscript.

**Funding:** Funded by national funds through FCT—Portuguese Science and Technology Foundation, within the project reference UIDB/04470/2020.

**Data Availability Statement:** The data of the current research is available from the corresponding author on request.

**Conflicts of Interest:** The authors declare no conflict of interest.

## Appendix A

Example of Variables and Results Values for the city of Ávila

| | MAX | MIN | MEAN | STD |
|---|---|---|---|---|
| PHASE I (WALKABILITY) | | | | |
| Facilities and services | 1000 | 264.78 | 706.77 | 191.61 |
| Accessibility | 1000 | 332.02 | 735.77 | 124.66 |
| Sidewalk | 1000 | 177.87 | 728.88 | 224.22 |
| Population density | 1000 | 0.08 | 7.21 | 42.28 |
| Green areas | 1000 | 161.11 | 739.93 | 224.42 |
| Urban trees | 1000 | 0 | 11.19 | 105.19 |
| WALKABILITY | 934.57 | 121.25 | 439.17 | 130.33 |
| PHASE II (WALKABILITY UNDER CLIMATE PRESSURE) | | | | |
| Temperature | 1000 | 751.91 | 916.20 | 40.94 |
| Solar radiation | 1000 | 0 | 898.84 | 93.02 |
| Shadows | 1000 | 0 | 85.68 | 148.48 |
| WALKABILITY 2020 | 834.23 | 82.71 | 405.97 | 86.77 |
| PHASE III (WALKABILITY UNDER FUTURE CLIMATE PRESSURE) | | | | |
| WALKABILITY 2030 A1B | 434.14 | 48 | 274.16 | 29.63 |
| WALKABILITY 2030 A2 | 431.31 | 40 | 272.01 | 29.42 |
| WALKABILITY 2050 A1B | 431.08 | 32 | 271.03 | 29.63 |
| WALKABILITY 2050 A2 | 429.12 | 25 | 268.95 | 29.42 |
| WALKABILITY 2100 A1B | 420.45 | 13 | 258.03 | 28.31 |
| WALKABILITY 2100 A2 | 419.43 | 0 | 256.89 | 29.42 |

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
