# Peer review of "Walkability under Climate Pressure: Application to Three UNESCO World Heritage Cities in Central Spain"

_land, doi:10.3390/land12050944_

Round 1

Reviewer 1 Report

The article has an interesting and useful topic from the point of view of localities whose economic engine is tourism. Unfortunately, I could not notice a clear connection, measured by any indicator, between walkability and tourism. Moreover, in the presentation of the 3 cities, there is no reference to tourism (for example, the size of the facilities, the number of tourists, etc.).

As such, the statement "... this work shows a  combination of the walkability index, within the tourism context of three World Heritage cities, linking not only walkability and tourism but also the potential affection due to climate change trends" (lines 107-109) does not have a clear correspondence with the analyzes carried out in the rest of the article.

My suggestion would be not to insist on the link with tourism if it cannot be evaluated with the help of some indicators.

Author Response

Dear Reviewer,

Many thanks for the constructive comments and suggestions.

We inform you that we considered them all, and based on such comments, improvements in themanuscript were made.

In fact, those improvements can be found in the manuscript body in blue text color.

The article has an interesting and useful topic from the point of view of localities whose economic engine is tourism. Unfortunately, I could not notice a clear connection, measured by any indicator, between walkability and tourism. Moreover, in the presentation of the 3 cities, there is no reference to tourism (for example, the size of the facilities, the number of tourists, etc.).

Based on the above-mentioned suggestions, we have created a new section regarding the study limitations and future research lines.

As such, the statement "... this work shows a combination of the walkability index, within the tourism context of three World Heritage cities, linking not only walkability and tourism but also the potential affection due to climate change trends" (lines 107-109) does not have a clear correspondence with the analyzes carried out in the rest of the article.

This specific sentence/stateme has been revised.

My suggestion would be not to insist on the link with tourism if it cannot be evaluated with the help of some indicators.

This issue has been revised.

Reviewer 2 Report

Dear  authors,

The article deals with a very important issue in the field of walkability under climatic pressure. The authors propose a method for predicting the effects of climate change by calculating a current walkability index based on OMS, lidar, and Sentinel-3 data and examining the walkability index under future climatic conditions. The article is well written, the working methodology is clearly defined, but I would have a few comments:
- The legend to the images should be clearer and more understandable, including textual explanations,
- The article includes images to visually illustrate the study, but it is recommended to include a short text explanation with each image.
- There are too many images spread over several pages and included in a single figure (e.g., Figure 4). I recommend combining them on one page or adding explanatory text if they are difficult to understand.

I recommend acceptance of the article after revision

Author Response

Dear reviewer,

Many thanks for the constructive comments and suggestions.

We inform you that we considered them all, and based on such comments, improvements in themanuscript were made.

In fact, those improvements can be found in the manuscript body in blue text color.

The article deals with a very important issue in the field of walkability under climatic pressure. Theauthors propose a method for predicting the effects of climate change by calculating a currentwalkability index based on OMS, lidar, and Sentinel-3 data and examining the walkability indexunder future climatic conditions. The article is well written, the working methodology is clearlydefined, but I would have a few comments:

- The legend to the images should be clearer and more understandable, including textual explanations,

Improved.

- The article includes images to visually illustrate the study, but it is recommended to include a shorttext explanation with each image.

Improved.

- There are too many images spread over several pages and included in a single figure (e.g., Figure

4). I recommend combining them on one page or adding explanatory text if they are difficult tounderstand.

We believe all the images have a high quality and are perfectly understandable by the reader.

I recommend acceptance of the article after revision

Reviewer 3 Report

Any work that brings a new methodology to sustainable habits in cities is exceptional. In this work, in a relatively simple way, the spaces that are sensitive for humans or will be in the future have been singled out, and that is the greatest quality of this work. What he lacks are data on tourists that prove that these are indeed tourist cities where it is important to increase walkability and reduce pollution. Also, some climate data are missing in order to get a complete picture of the climate of the places that were investigated. The biggest drawback of the work is the short time series given for the air temperature. In order to apply the given methodology, each variable should be as representative as possible, including temperature. I asked the authors to try to edit that part. Comments are in the attached version.

Author Response

Dear Reviewer,

Many thanks for the constructive comments and suggestions.

We inform you that we considered them all, and based on such comments, improvements in the manuscript were made.

In fact, those improvements can be found in the manuscript body in blue text color.

Any work that brings a new methodology to sustainable habits in cities is exceptional. In this work, in a relatively simple way, the spaces that are sensitive for humans or will be in the future have been singled out, and that is the greatest quality of this work. What he lacks are data on tourists that prove that these are indeed tourist cities where it is important to increase walkability and reduce pollution.

Also, some climate data are missing in order to get a complete picture of the climate of the places that were investigated. The biggest drawback of the work is the short time series given for the airt emperature. In order to apply the given methodology, each variable should be as representative as possible, including temperature. I asked the authors to try to edit that part. Comments are in thea ttached version.

Based on the above-mentioned suggestions as well as on the attached pdf created by the reviewer, we have developed a new section regarding the study limitations and future research lines.